# Effect of Climate on Bacterial and Archaeal Diversity of Moroccan Marine Microbiota

**DOI:** 10.3390/microorganisms10081622

**Published:** 2022-08-10

**Authors:** Yousra Sbaoui, Abdelkarim Ezaouine, Marwene Toumi, Rózsa Farkas, Mouad Ait Kbaich, Mouna Habbane, Sara El Mouttaqui, Fatem Zahra Kadiri, Mariame El Messal, Erika Tóth, Faiza Bennis, Fatima Chegdani

**Affiliations:** 1Department of Biology, Immunology and Biodiversity Laboratory, Faculty of Sciences Ain Chock, Hassan II University of Casablanca, Casablanca 20000, Morocco; 2Department of Microbiology, Eötvös Loránd University, 1117 Budapest, Hungary; 3Departement of Molecular Medicine, The Scripps Research Institute, La Jolla, CA 92037, USA; 4Departamento de Bioquímica, Biología Molecular y Celular, Universidad de Zaragoza, C/Miguel Servet, 177, 50013 Zaragoza, Spain; 5Laboratoire Biologie et Santé, Faculté des Sciences Ben M’Sick, Hassan II University of Casablanca, Sidi Othman, Casablanca 20670, Morocco; 6Department of Biology, Immunophysiopathology, Biochemistry and Biotechnology Laboratory, Faculty of Science Ain Chock, Hassan II University of Casablanca, Casablanca 20000, Morocco

**Keywords:** Moroccan marine microbiota, climate, Metagenomics, diversity, NGS, Mothur, microbiome analyst, R software

## Abstract

The Moroccan coast is characterized by a diversity of climate, reflecting a great richness and diversity of fauna and flora. By this, marine microbiota plays a fundamental role in many biogeochemical processes, environmental modifications, and responses to temperature changes. To date, no exploration by high-throughput techniques has been carried out on the characterization of the Moroccan marine microbiota. The objective of this work is to study the diversity and metabolic functions of MMM from the Moroccan coast (Atlantic and Mediterranean) according to the water source (WS) and the type of climate (CT) using the approach high-throughput sequencing of the 16SrRNA gene. Four water samples of twelve sampling sites from the four major climates along the Moroccan coastline were collected, and prokaryotic DNA was extracted. V4 region of 16S rRNA gene was amplified, and the product PCR was sequenced by Illumina Miseq. The β-diversity and α-diversity indices were determined to assess the species richness and evenness. The obtained results were analyzed by Mothur and R software. A total of twenty-eight Bacterial phyla and twelve Archaea were identified from the samples. Proteobacteria, Bacteroidetes, and Cyanobacteria are the three key bacterial phyla, and the Archaeal phyla identified are: *Euryarchaeota*, *Nanoarchaeaeota*, *Crenarchaeota, Hydrothermarchaeota*, *Asgardaeota*, *Diapherotrites*, and *Thaumarchaeota* in the Moroccan coastline and the four climates studied. The whole phylum are involved in marine biogeochemical cycles, and through their functions they participate in the homeostasis of the ocean in the presence of pollutants or stressful biotic and abiotic factors. In conclusion, the obtained results reported sufficient deepness of sequencing to cover the majority of Archaeal and Bacterial genera in each site. We noticed a strong difference in microbiota diversity, abundance, and taxonomy inter- and intra-climates and water source without significant differences in function. To better explore this diversity, other omic approaches can be applied such as the metagenomic shotgun, and transcriptomic approaches allowing a better characterization of the Moroccan marine microbiota and to understand the mechanisms of its adaptation and its impacts in/on the ecosystem.

## 1. Introduction

Microbes are ubiquitous in the ocean environment and play fundamental roles in many biogeochemical processes, including carbon and nutrient cycling [1]. In each milliliter of water there are 10^4^ to 10^6^ cells, involved in the production of biomass, renewal, genetic diversity and environmental complexity [2]. The ocean microbiome with its microbiota participates in strong environmental changes and responds to temperature associated with climate change, carbon chemistry, nutrient, oxygen content, and variation in stratification and ocean currents [3].

The progression of DNA sequencing techniques to study these microbial communities by different molecular approaches (Metagenomics, Metatranscriptomics, Amplicon sequencing) has greatly contributed to obtaining information related to the genomes and metagenomes of marine microbes in order to study their structural patterns, their diversity, and their functional potential [4], with a combination to environmental data measured in situ, to provide a global context and minimize potential confounding factors. 16S rRNA-based marker gene sequencing is widely used to study and characterize the taxonomic composition and phylogenetic diversity of these microbial communities, followed by downstream statistical, visual, and functional analysis to better understand the communities [5]. However, unlike terrestrial genomics programs, the proportion of unknown protein families is greater in marine organisms because the oceans have been less examined. It is therefore necessary to characterize genes/proteins to interpret marine genomic and metagenomic data. This is to understand the fundamental mechanisms of the marine environment, such as interactions between organisms or the adaptation of populations to environmental changes [4].

However, despite our knowledge about marine microbes and their global importance and recent significant advances in marine microbiome research, the diversity of the Moroccan marine microbiota (MMM) has never been characterized and analyzed in terms of Moroccan Littoral (ML) and climate type (CT), particularly since Morocco is a very diversified climatic country, which brings into consideration some fundamental unanswered questions that serve as active drivers for current research: How many microbes exist, and what are their types? What are their functional roles? How are they distributed along the ML? How do they adapt to the various environmental conditions in each climate and how will they respond to future environmental changes?

To answer these questions, in this study we are interested in investigating the diversity, abundance, and metabolic functions of MMM from the Moroccan coastline (Atlantic and Mediterranean) in function of water source (WS) and climate type (CT). The diversity and taxonomy of MMM will be studied in the surface microlayer through the 16S rRNA amplicon sequencing approach in four major climate types (Mediterranean, Semi-arid, Arid, and Saharian). Raw reads processing and taxonomic assignment will be performed using the Mothur pipeline and the MicrobiomeAnalyst package in R Studio.

## 2. Materials and Methods

### 2.1. Selection and Description of Sampling Sites

The two Moroccan Atlantic and Mediterranean coasts have been targeted according to 4 Moroccan climates: Mediterranean, Semi-Arid, Arid, and Saharian (http://koeppen-geiger.vu-wien.ac.at/present.htm, accessed on 20 December 2019). Water sampling was carried out at the level of the surface microlayer (0–200 µm) in the Mediterranean (35°48′27.5″ N; 5°21′03.5″ W), Arid (33°52′12.3″ N; 7°03′26.5″ W), Semi-Arid (32°44′43.8″ N; 9°02′03.9″ W), and Saharian (23°45′36.0″ N; 15°55′12.0″ W) climates (Figure 1a,c). The sites (Figure 1b) are selected according to the NM 03.7.199 standard, used to classify the bathing waters of Moroccan beaches on A, B, C, and D. The classification is done by a monitoring program (Version 2019: https://www.environnement.gov.ma/fr/zones-cotieres?id=222, accessed on 15 June 2019) for the microbiological and physico-chemical quality of bathing water, developed jointly by the Ministry of Equipment, Transport, Logistics, and Water and the Secretariat of State for Sustainable Development. Thus, category A sites, reflecting good quality water, were chosen.

### 2.2. Sampling

Water samples were collected under sterile conditions and as representative as possible of the environment from which they were taken in clean and sterile glass vials according to the ISO 19458:2006 standard [6]. Three sampling points (upstream, downstream, and in the middle) were chosen in each climate following the marine current (east–west). Sampling was done in triplicate at low tide in an area on the shoreline in permanent exchange with offshore water in the period between 2019 and 2020. The in situ determination of pH, temperature, dissolved oxygen, and salinity were measured using a portable multi-parameter Eutech Instruments (CyberScan). Samples were kept at 4 °C during transport and processed immediately upon arrival at the laboratory. A sample handling sheet for a sea water sample is available in the Appendix A. In addition, to understand the major mechanisms linking microbiome and climate, data on chemistry and atmospheric particles were taken from the TaraMicobiome mission (https://tara.nullschool.net/?microbiomes), Table 1.

### 2.3. Sample Preparation and DNA Extraction

The 3 points sampled in each CT with the triplicates were pooled (3L × 3) to testify on the total of autochthonous diversity and then filtered using a sterile mixed nitrocellulose filter with a porosity of 0.22 µm (MF-Millipore GSWP04700, Billerica, MA, USA). Total DNA for each climate was extracted using a DNeasy^®^ PowerSoil^®^ DNA isolation kit (QIAGEN, Hilden, Germany) according to the manufacturer’s instructions. Mechanical disruption of the cells was performed by shaking at 30 Hz for 2 min using a Retsch Mixer Mill MM400 (Retsch, Haan, Germany).

### 2.4. PCR Amplification

For PCR reactions, an amount of 5 µL with a concentration between 1 to 5 ng/µL of template DNA was used to amplify the 16SrRNA V4 region following the protocol in Table 2. PCR products were quantified using a Qubit meter (Invitrogen Life Technologies, CA, USA), and a minimum concentration of 4 ng/μL for a final volume of 50 μL was set before sending to sequencing.

### 2.5. Illumina Sequencing and Data Analysis

#### 2.5.1. 16S rRNA Gene Sequencing

Relative abundance of Bacterial and Archaeal phyla in water samples was estimated by sequencing the PCR amplicons targeting V4 16SrRNA region using the Illumina standard MiSeq v2 chemistry platform, Michigan State University. https://support.illumina.com/documents/documentation/chemistry_documentation/16s/16s-metagenomic-library-prep-guide-15044223-b.pdf, accessed on 5 March 2022.

#### 2.5.2. Sequence Analysis

Sequence analysis was done by the Mothur pipeline (version1.46.1, Mothur software, https://mothur.org/) [9], and the fastq (paired-end reads) forward and reverse files obtained from the Illumina sequencer were processed and analyzed. The contigs were obtained using the “make.contigs” command with a deltaq value of 10, in order to keep the sequences with high quality scores. Then, in order to keep only the sequences filling the expected length and number of polymers and ambiguous bases, the command “screen.seqs” was used. Sequences were aligned to the Silva database (version 138_1, silva.nr.align database, https://www.arb-silva.de/) [10], and unaligned sequences and columns containing only “.” were removed using the “screen.seqs” and “filter.seqs” commands, based on the position of the Archaeal and Bacterial primers in the 16S rRNA gene [11].

To remove sequences likely due to Illumina sequencing errors, the “pre.cluster” command was used. Chimeric sequences were removed using the UCHIME algorithm (version11, uchime algorithm, https://www.drive5.com/usearch/manual/uchime_algo.html) [12]. Only abundant sequences were retained using the “split.abund” command, which split the sequences into two groups, with a threshold value equal to 1. Taxonomic classification of sequences was performed using the Silva database (version: 138_1, silva.nr.tax; https://www.arb-silva.de/) and the non-bacterial, archaeal sequences were removed from the analyses based on the results of the taxonomic classification [10]. OTUs (Operational Taxonomic Units) were calculated using a distance matrix with distances greater than 0.15 obtained using the “dist.seqs” command and later the cluster commands to assign sequences to OTUs, and finally consensus taxa were determined using the “classifier.otu”. At the end, the data were normalized using the “sub.sample”, and the “rarefaction.single” and “summary. Single” were used to calculate the rarefaction curve data and diversity index values [8].

#### 2.5.3. Taxonomic Profiling of the Data

The study of microbial community diversity (Archaeal and Bacterial) based on microbial ecology methods was performed with MicrobiomeAnalystR (version: 4.0.2; microbiomeanalyst software, https://www.microbiomeanalyst.ca). Taxonomic abundance profiles in function of CT and WS were visualized at the genus level by a stacked bar diagram and at all levels by an Interactive Krona chart [13] in the Silva-ngs platform (version: 1.9.5/1.4.6, silvangs, https://ngs.arb-silva.de/silvangs/). To establish whether α-diversity differs between CT samples and between WS, reads were transformed and low-abundance OTU were filtered from the datasets. The Chao1 index was used to estimate OTU richness by identifying rare OTUs, while the Shannon index was used to estimate both OTU richness and regularity.

On the other side, to compare the differences in the microbiome between CT and WS samples based on distance or dissimilarity measures, a dissimilarity matrix was generated from the log-transformed sequence data, and the ordination of the plots was visualized using 2D principal coordinate analysis (PCoA). The Bray–Curtis distance matrix was used to visualize differences in microbiome composition. Identification of taxa or core features that remain unchanged in composition across CT and WS was done through the CoreMicrobiomeAnalysis function based on prevalence and relative abundance at the genus level.

#### 2.5.4. Functional Profiling

Prediction of microbial community functions were evaluated using the R package “Tax4Fun (version: 3.10, tax4fun software, http://tax4fun.gobics.de/). The functions of the 16S rRNA marker genes were linked to orthologs (KOs) in the Kyoto Encyclopedia of Genes and Genomes (KEGG) SILVA database using the MoP-Pro approach [13,14], to determine the relative abundance of the predictive function genes for each MMM sample. The top 11 KO were then selected and plotted on bar graphs to associate relative gene abundance to CT and WS. Orthologs that were not linked to a specified level-one KEGG pathway were excluded from the analysis, and the next most abundant gene was substituted. Functions were combined on the basis of level one as defined by KEGG pathway output.

#### 2.5.5. Comparative Analysis

The differential abundance of OTU among samples from different CT and WS was performed on variance-stabilized data that were agglomerated at the genus level by the DESeq2 package (version 1.34.0, DESeq2 software, https://bioconductor.org/packages/DESeq2/), using negative binomial generalized linear models to estimate dispersion and log fold changes. To compare differences between communities, a hierarchical clustering was established based on the distance measure between clusters. The distance between genera is calculated using the Euclidean method and clustered using the Ward algorithm.

Since the microbiome data are compositional, analyzing the differences between samples using conventional statistical tests can lead to false hypotheses [15]. To further support the results, the pattern search function of MicrobiomeAnalyst was used by specifying the coastline (Atlantic vs. Mediterranean) as a feature. The Hollow Correlations for Composition Data (SparCC) command was used as a measure of taxon distance between features to identify correlations of individual taxa with groups.

#### 2.5.6. Statistical Analysis

Statistical analyses were thus performed on MicrobiomeAnalyst software, in R studio (version 4.0.2, Rstudio software, http://www.r-project.org), and the significance level applied for all statistical tests was 5% (*p* < 0.05). Statistical significance was inferred using the Mann–Whitney/Kruskal–Wallis method for α-diversity and classical univariate comparison analysis, whereas permutational MANOVA (PERMANOVA) was used to test statistical significance of dissimilarity measures. For function prediction, Welch’s two-sample *t*-tests were followed. Detection of OTU that expressed differential abundances between different CT was done by group comparison using a parametric test (ANOVA) with log2FC > 2 and *p*-value < 0.05 adjusted by FDR method.

### 2.6. Sequence Availability

Sequence reads were deposited in the NCBI SRA database and are accessible through the BioProject ID: PRJNA841938 and BioSample ID: SAMN28631598 for Saharian, SAMN28631599 for Arid, SAMN28631600 for Semi_arid, and SAMN28631601 for Mediterranean.

## 3. Results

### 3.1. Physical and Chemical Parameters of Water Samples

The physical and chemical in situ parameters of different sampling sites measured during sampling mission are presented in Table 3 and accompanied with spatial data from Tara microbiome mission in Figure 2.

### 3.2. Taxonomic Composition Based on Total Community

V4 region sequencing in the 16S rRNA coding gene, was performed to study the structural composition of the marine microbiome and the relative abundance of the different components according to CT and WS. The results of this sequencing gave between 100–130 K raw reads per sample. Of these, 60%–76% passed quality control. These processed reads varied in size from 370–480 bp with an average sequence size of 436 bp for each sample. Sample rarefaction curves (Appendix A) showed that the sequencing depth was sufficient to recover the majority of bacterial and archaeal taxa. The samples uniformly showed the presence of Superkingdom (SK) Bacteria and Archaea with different proportions, as the main constituent of the MM in each CT. Histogram and Krona chart (Figure 3 and Figure 4) show the major taxonomic groups for each sample according to the whole community of prokaryotes. A total of 28 and 12 Bacterial and Archaeal phyla, respectively, were found by DNA sequence homology with reference genomes from the Silva database.

The major Bacterial phyla present in the four samples were Actinobacteria, *Bacteroidetes, Cyanobacteria*, *Patescibacteria*, *Planctomycetes*, *Proteobacteria*, and *Verrucomicrobia* (Appendix A) and for Archaea, the major phyla are Euryarchaeota, Nanoarchaeaeota, and *Thaumarchaeota* (Appendix A). *Marinimicrobia_(SAR406_clade)* and *Epsilonbacteraeota* dominate the Saharian, Arid, and Mediterranean climates. *Firmicutes* and *Crenarchaeota* for Bacteria and Archaea, respectively, dominate the Saharian, Arid, and Semi-arid climates. *Fusobacteria* dominate the Saharian, Arid, and Mediterranean climates. *Omnitrophicaeota* and *Hydrothermarchaeota* for Bacteria and Archaea, respectively, dominate the Saharian climates. *Diapherotrites* for Archaea dominates the Arid climate, and *Asgardaeota* dominates the Semi_arid climate. However, a large proportion of reads in each sample could not be assigned to any taxonomic rank below the domain and were labeled as unclassified_Bacteria and Archaea (Appendix A).

The dominant genera belong unambiguously to the three phyla *Proteobacteria, Bacteroidetes*, and *Cyanobacteria* (Figure 4). *Proteobacteria* present in the Mediterranean climate showed a high ratio of *Amylibacter* (16%); in the arid climate, we find a high ratio of *Pseudoalteromonas* (59%) and *Psychrobacter* (13%). For the Saharian climate, we find *Pseudoalteromonas* (32%), and for the semi_arid climate, we have *Pseudoalteromonas* (39%).

For *Bacteroidetes*, in the Mediterranean climate we find *Polaribacter* (14%), and at the level of *Cyanobacteria* we find *Chloroplast* (76%). In the arid climate, there are *Sediminibacterium* (77%), and for *Cyanobacteria*, we find *Chloroplast* (67%). In the Saharian climate, we find *NS5_marine_group* (21%), and for *Cyanobacteria* we find *Chloroplast* (84%). For *Cyanobacteria,* we find *Chloroplast* (80%).

### 3.3. Alpha and Beta Diversity

#### 3.3.1. Alpha diversity

Figure 5 shows the α-diversity observed among the climate samples. The samples could be sequenced at a variable depth range from 1 × 10^5^ to 2 × 10^5^ and showed variable regularity and richness of microbial diversity among them, even though Archaea and Bacteria were associated with the same CT each time. According to Shannon and Chao1 indexes (Figure 5a,c), at the level of the Atlantic site, the Saharian, and the Semi-arid, we noticed the highest Bacterial diversity while the Arid climate has the lowest Bacterial diversity. At the level of the Mediterranean site, the Mediterranean climate expressed good Bacterial diversity. Nevertheless, the Semi-arid climate in the Atlantic coast and Mediterranean climate in Mediterranean coast expressed a great richness of Achaean communities while the Arid climate in the Atlantic coast shows the lowest richness. To evaluate the microbial diversity between the two sources studied, a peer comparison (Figure 5b,d) of the abundances between the Atlantic and Mediterranean seaboard was carried out. The whiskers in the whisker box represent the range of minimum and maximum α-diversity values within a population, excluding outliers that occur as single points, based on Kruskal–Wallace, Shannon, and Chao1 *p* < 0.05 results. Collectively, these data indicate that the diversity of the MMM varies with CT and WS.

#### 3.3.2. Beta Diversity

The β-diversity among the samples was analyzed by PCA (Principal Component Analysis) (Figure 6). Results shows that each sample is unique and has varying diversity and abundance of OTUs relative to each other. The PC1 axis was more informative than PC2 on β-diversity. Samples were split into two sources based on CT and scoring. In Figure 6, the samples were shown with a different color to indicate the source they belong to. These two sources are the Atlantic and the Mediterranean.

Regarding the Bacterial communities, the Saharian and semi-Arid samples at the Atlantic level form a group, indicating that they have a close and common diversity. A similar trend was observed for the Archaean communities for the two climates Sahara_arch and Mediterannean_arch, associated, respectively, with the Atlantic and Mediterranean sites. In addition, the statistics of the graph indicated a significant difference in β−diversity between the two coasts and the four types of climates studied (*p* < 0.05).

### 3.4. Hierarchically Clustered Heat Map

The heat map in Figure 7, was constructed to represent the relative abundance of various Bacterial and Archaeal genera in the four samples. It shows that most genera occur at low abundance. The samples were found to have 50 genera in common, and these occurred in varying proportions between sources and samples. The Atlantic and Mediterranean climates were both associated with Bacteria and Archaea; however, in the Saharian and Semi−arid climates, there was a high abundance of Bacterial communities at the expense of Archaea, while in the Arid and Mediterranean climates comparable proportions were found. Moreover, by the statistical test, we have noticed a significant difference in abundance between Archaeal and Bacterial communities in function of climate and score (Kruskal–Wallis, FDR corrected *p* < 0.05). The distance between features (genera) is calculated using the Euclidean method, and they are clustered using Ward’s algorithm.

### 3.5. Core Microbiome Analysis

Despite the changes in diversity detected by the α- and β-diversity study in function of CT and WS from both Moroccan coasts, a substantial fraction of the total OTU was shared between all samples (Figure 8). This large fraction of shared taxa constitutes the core microbiome of the Moroccan marine microbiota, resilient to stressful and fluctuating conditions of the biotic and abiotic parameters of the host environment.

The core microbiome of the present study, which remains unchanged across the four CT and two coasts studied based on prevalence and relative abundance (Figure 8) included 1 unclassified phylum (*Uncultured_bacterium*), 3 unclassified families (*Nitrosopumilaceae_unclassified, Rhodobacteraceae_unclassified,* and *Vibrionaceae_unclassified*), and 16 genera (*Candidatus_Nitrosopulmius, Marine_Group_II_ge, Pseudoalteromonas, Bacteroides, Clade_la, RC9_gut_group, Amylibacter, Vibrio, Candidatus_Nitrosopelagius, Incertae_Sedis, Cobetia, Aurantivigra, NS5_marine_group, Glacicola, Coprococcus*, and *Candidatus_Actinomarina*).

### 3.6. Correlation Analysis

To further strengthen the statistically significant differences in abundance between the two coasts described, a compositional method combining pattern correlation (Figure 9) and thermal map analysis was used to search for distance patterns to identify correlations of taxa at the genus level present in the Atlantic with those in the Mediterranean. This correlation was done by a comparison of relative abundance using linear discriminant analysis effect size (LEfSe).

The results of these correlation are shown in Table 4. Similar observations were observed in the heat map where the genera *Polaribacter*, *Yoonia_Lokatanella*, *Amylibacter, Planktomarina, SUP05_cluster, SAR86_clade_ge,* and *HIMB11* show the highest presence in the Mediterranean, and *Alistipes, Solibacillus, Bacteroides*, *Coprococcus*, and *Incertae_Sedis* in the Atlantic.

### 3.7. Differential Abundance Analysis

Using a group comparison parametric test (ANOVA) on differential abundances of a factor log2FC > 2 and *p*-value < 0.05 of OTU, there was a revelation of significant fluctuations in the abundances of several genera (Table 5). A total of 26 genera with significant characteristics showed differential abundance by CT and WS. The *p*-values are adjusted by the FDR method.

### 3.8. Functional Profiling and Metabolic Network Visualization

Metabolic function analysis was used to predict the metabolism of microbial communities associated with the four CT and WS. The prediction is based on sequencing data of amplicons of closely related taxa. The microbiomes of the different climates showed similar proportions of KEGG pathways associated with amino acid metabolism, secondary metabolite biosynthesis, carbohydrate metabolism, energy metabolism, glycan metabolism and biosynthesis, lipid metabolism, cofactor and vitamin metabolism, terpenoid and polyketide metabolism, nucleotide metabolism, and biodegradation of xenobiotics (Figure 10a). On the other hand, the specific analysis of the metabolic profiles in terms of statistically significant differences in pathway abundances (Figure 10b) between the different climates through an ANOVA (*p* < 0.05) followed by *Holm*–*Bonferroni* correction (FWER) showed no significant differences.

## 4. Discussion

The objective of our study was to investigate the diversity and metabolic functions of Moroccan marine microbiota (Atlantic and Mediterranean) according to sampling sites (WS) and type of climate (CT). Water samples from the surface microlayer of four major CT (Mediterranean, Semi-arid, Arid, and Saharian) were collected, and the extracted DNA was used to generate 16S rRNA gene libraries. Processing of the raw reads and taxonomic assignment was performed using the Mothur pipeline and CSV files with relative abundance for Archaea and Bacteria independently processed by the MicrobiomeAnalyst package in R Studio. The taxonomic profile of MMM obtained by linear discriminant analysis suggests a correlation between CT and WS (R^2^ = 0.95).

The different CT were dominated by typical genera of *Proteobacteria* including (Mediterranean = *Amylibacter, Clade_la,* and *SAR86_clade;* Saharian = *Pseudoalteromonas, Amylibacter,* and *Vibrio;* Semi_arid = *Pseudoalteromonas, SAR116_clade,* and *Oceaniserpentilla*; Arid = *Pseudoalteromonas, Psychrobacter,* and *Cobetia*) and for *bacteroidetes* genera including (Mediterranean = *Polaribacter, Aurantivigra,* and *N5_marine_group*; Saharian = *NS5_marine_group, Polaribacter,* and *Tenacibaculum;* Semi_arid = *NS5_marine_group, NS4_marine_group,* and *Aurantivirga;* Arid = *Sediminibacterium* and *Terrimonas*) both in terms of relative abundance and taxonomic richness. It has been shown that microorganisms play a crucial role in climate regulation through their varied responses combined to different abiotic parameters [16].

Our results suggest that each climate expresses a specific microbiota composition influenced by the availability of nutrients and physicochemical parameters. This is relatively true because along the Moroccan coastline, each climate has expressed a specific microbiota composition influenced by the stratification of the oceans and the diversity of micro-organisms [17].

The major genera of *Proteobacteria* and *Bacteroidetes* are gram-negative bacteria very influenced by temperature changes in different climates [18]. Additionally, a decrease in fungal abundance and modification of gene expression of some Archaea and Bacteria impact diversity of these two groups [19]. However, photosynthetic *Cyanobacterial* taxa such as *Chloroplast, Prochlorococcus,* and *Synechococcus* were detected in all climate samples and contributed about 10% of the abundance, which is in agreement with the reports of the climate institute that describes the effect of climate on the proliferation of *Cyanobacteria* especially in warm waters [20]. This suggests the role of *cyanobacteria* in ecosystem development, microbiome, and biogeochemical cycles [21].

Concerning Shannon’s α-diversity index, samples from the Saharian, Semi-Arid, and Mediterranean climates showed higher diversity than samples from the Arid climate. However, samples from both the Atlantic and Mediterranean coasts showed good richness. Collectively, these data indicate that MMM diversity varies with WS and CT. These results are consistent with those of Sebastien et al., 2021 where they report the effect of environmental gradients and physical barriers on the spatial determination and variation of vertical and horizontal distribution of microbial communities between the Atlantic and Mediterranean [22]. Furthermore, the PCoA plot indicated a significant difference in beta diversity between the two coasts and the four CT studied (*p* < 0.05). This is in agreement with several studies that have reported vertical stratification of microbial taxa and viruses based on physicochemical changes, such as light, temperature, and nutrients [14,23,24,25]. Considering the vertical and horizontal stratification, we characterized the taxonomic and functional richness, dissimilarity between samples (β-diversity), total cell abundance, and potential growth rates in the four CT studied. Our results revealed an increase in taxonomic richness with CT and WS (Figure 6 and Figure 7).

For a better understanding of the underlying mechanisms, we investigated whether the samples were more similar within CT than between them, focusing only on the surface microlayer. The dendrogram results showed that the samples were clustered more efficiently based on WS than CT, and the heat map showed that the samples were found to have 50 genera in common and these occurred in varying proportions between WS and CT (Figure 7 and Figure 8) In addition, differential abundance analysis using DESeq2 at the OTU and genus levels revealed significant fluctuations in the abundances of 26 genera by CT and WS. Combining these results with a previous work, it was found that there is a strong impact of light and temperature on the biogeographic distribution of different microbial taxa [26]. Therefore, if environmental selection explained biogeographic patterns, we would expect environmental factors to be correlated with community similarity, which was validated by the β-diversity results. However, for whole-community assemblages, expectations are less clear. In a large-scale meta-analysis, salinity was suggested as the main determinant in many ecosystems (including oceans), overcoming the influence of temperature especially since the Mediterranean is more saline than the Atlantic [8,27,28,29].

The study of metabolic functions between different CT and WS in the surface microlayer by similar enrichments of KEGG pathways suggested the involvement of different metabolic pathways with amino acid metabolism, secondary metabolite biosynthesis, carbohydrate metabolism, energy metabolism, glycan metabolism and biosynthesis, lipid metabolism, cofactor metabolism and vitamins, metabolism of terpenoids and polyketides, nucleotide metabolism, and biodegradation metabolism of xenobiotics.

These functions are consistent with the fact that in the surface microlayer, energy production is a strong process requiring different sources and substrates including carbohydrates, lipids, and amino acids to respond to different stress factors and adapt to climate change [30]. The movements of microbes in a particular climate require energy to activate flagellar assembly mechanisms and chemotaxis [31]. In contrast, motility has been shown to reduce grazing mortality in planktonic bacteria [32]. Furthermore, these motility traits are potentially of great use for ocean bacteria to change areas and exploit new niches depending on physico-chemical and temperature parameters, which may lead to diversification of microbial populations adapted to different climates [32]. In the future, a thorough exploitation through transcriptomic data will allow to differentiate the active biomass from the dead biomass and if there are mechanisms of adaptation or gene transfers by horizontal and vertical mechanisms in the inner ocean.

## 5. Conclusions

The present study was designed to determine, for the first time, the effect of climate on the distribution of Moroccan Marine Microbiota along the Moroccan coastline using a combination of high-throughput sequencing and post-sequencing analysis of 16S rRNA, and we investigated how the MMM differs from one climate to another. As a result, we reported a strong difference in microbiota diversity, abundance, and taxonomy between CT and WS without significant differences in function.

Today, we live under the greenhouse effect as a result of climate change, and the study of the effect of climate on the diversity of marine microbiota gives long-term predictions of these effects. The idea is that the characteristics of one climate will shape the future of another climate, i.e., from one year to the next there is an increase in temperature, a decrease in precipitation, and an increase in ocean levels, so that one climate (e.g., Mediterranean) will exceed the arid climate by 2 °C in temperature; the specific composition and diversity will change if we go from 25 to 27 °C, but these data remain only predictions for the time being that we must validate by other approaches in combination with other ecological descriptors.

For in-depth use, the transcriptomic data will make it possible to understand the mechanisms of adaptation by horizontal or vertical gene transfer in the ecosystem and their impact on marine microbiotic diversity.

## Figures and Tables

**Figure 1 microorganisms-10-01622-f001:**
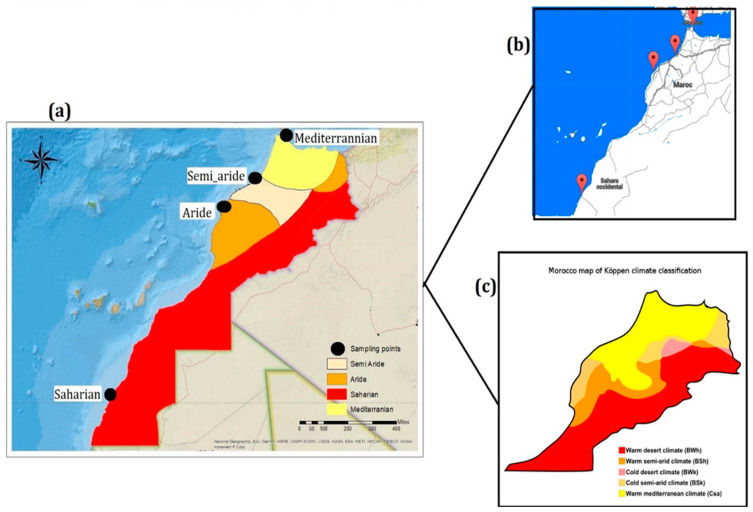
Sampling map of the MMM according to CT and microbiological quality of surface microlayer water. (**a**) Map of Morocco with the 4 dominant climate types (Beige = Semi_Arid; Orange = Arid; Yellow = Mediterranean; Red = Saharian; Black circle = Selected sites). (**b**) Geographic map with the location of the different sampling points. (**c**) Morocco map of Koppen climate classification.

**Figure 2 microorganisms-10-01622-f002:**
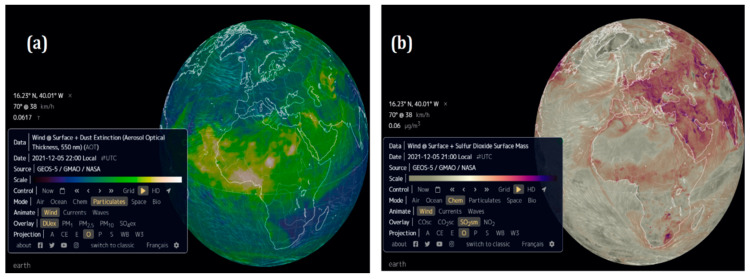
Representation of space data on atmospheric particles (**a**) and chemistry (**b**) shared by the Tara consortium, TaraMicobiome mission. Source (GEO-5/GMAO/NASA; https://earth.nullschool.net/).

**Figure 3 microorganisms-10-01622-f003:**
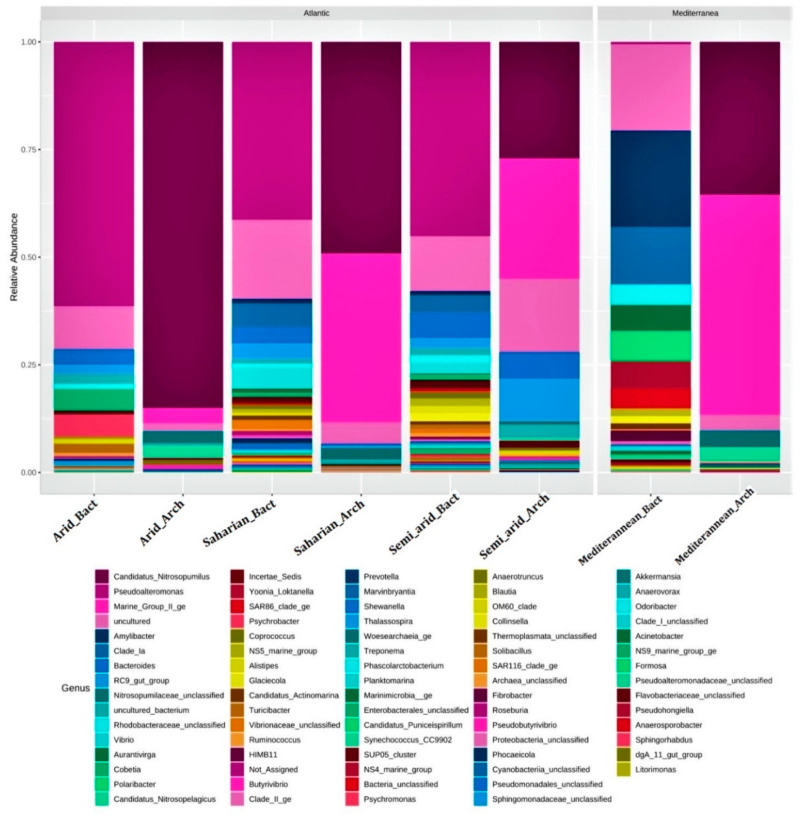
Histogram representing the distribution of major taxonomic groups for Archaea and Bacteria at the genus level in different types of climate. Bars are representing the relative abundance of taxonomic groups associated with MMM and revealed by metagenome sequencing. The color of each slice is based on sequence abundance, not on phylogenetic relatedness. This information is based on the Silva database used by Mothur to generate a features table. Slices with no taxonomic affiliation also include several taxonomic groups with a low number of sequences.

**Figure 4 microorganisms-10-01622-f004:**
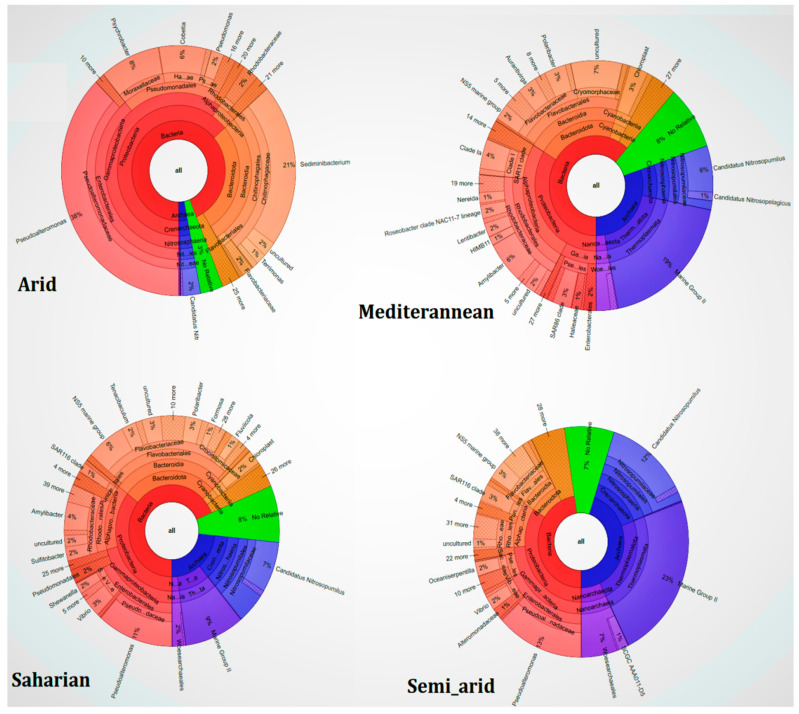
Krona map representing the taxonomic composition of the entire archaeal and bacterial community associated with MMM and CT, revealed by metagenome sequencing. The color of each slice is based on sequence abundance, not on phylogenetic relatedness. This information is based on the Silva database used by Mothur to generate the Krona chart. Slices with no taxonomic affiliation also include several taxonomic groups with a low number of sequences.

**Figure 5 microorganisms-10-01622-f005:**
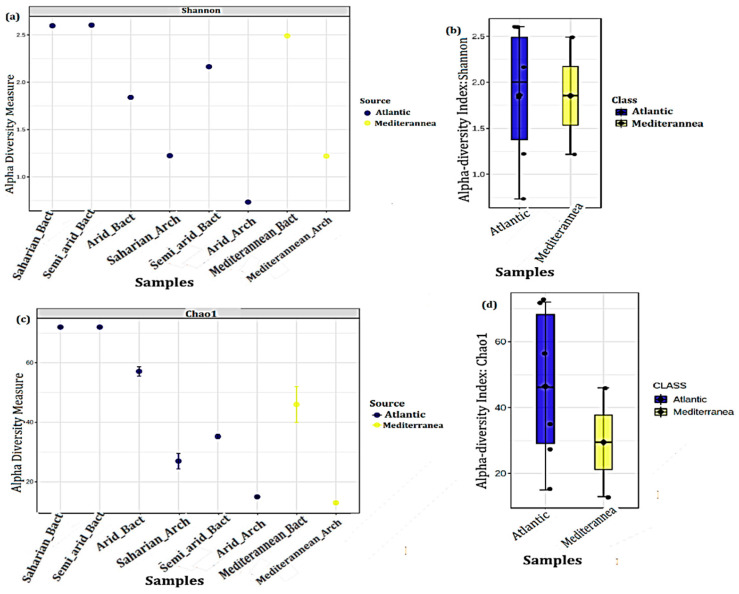
α-diversity of the four climate samples. Samples are plotted on the *X*-axis, their diversity on the *Y*-axis, and they show varying richness and evenness of microbial diversity based on the Shannon and Chao-1 index (**a**,**c**). Each sample is colored according to the source (metadata). The statistical significance (**b**,**d**) of the grouping based on source (Atlantic = Blue and Mediterranean = Yellow) is also estimated using the Kruskal–Wallace test with a *p*-value < 0.05.

**Figure 6 microorganisms-10-01622-f006:**
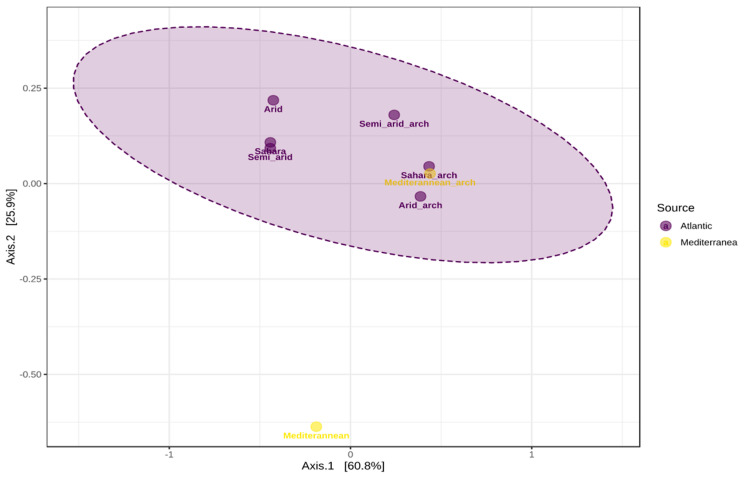
PCA of climate samples. Samples with different diversity were not clustered. Results are presented as 2D ordination plots based on principal coordinate analysis (PCoA). The corresponing statistical significance is assessed using permutational multivariate analysis of variance (PERMANOVA). Samples displayed on the PCoA plots are colored according to the metadata.

**Figure 7 microorganisms-10-01622-f007:**
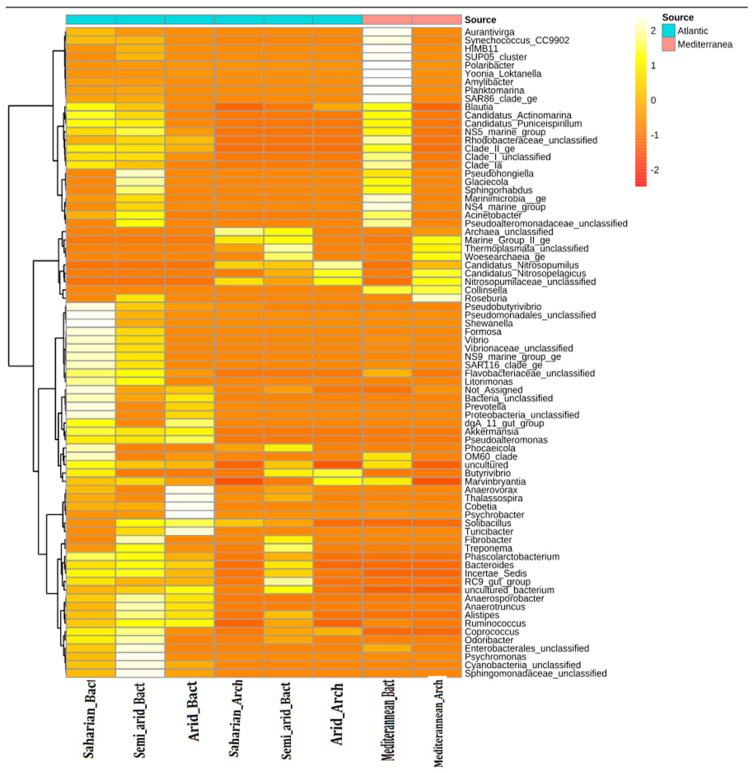
Hierarchically clustered heat map showing genus-level taxonomic abundance variation for Bacterial and Archaeal communities in climate samples as influenced by source. Feature count abundances are scaled to allow meaningful comparison between them. The heat map is co−or-coded by intensity based on taxonomic abundances. The feature clustering pattern is presented by a dendrogram on the left side of the heat map.

**Figure 8 microorganisms-10-01622-f008:**
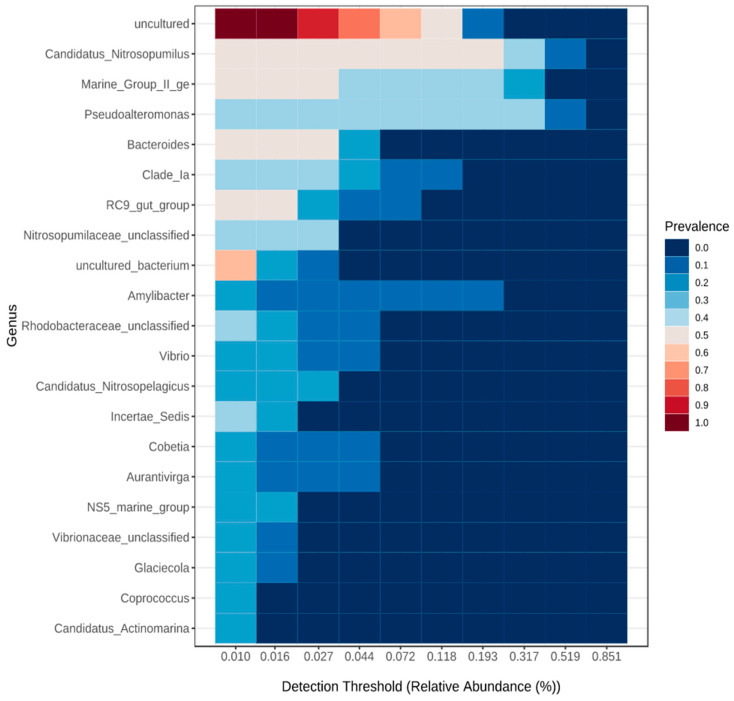
Heat map representing the central microbiome at the genus level using a detection threshold for prevalence of 10% and 0.1% for relative abundance.

**Figure 9 microorganisms-10-01622-f009:**
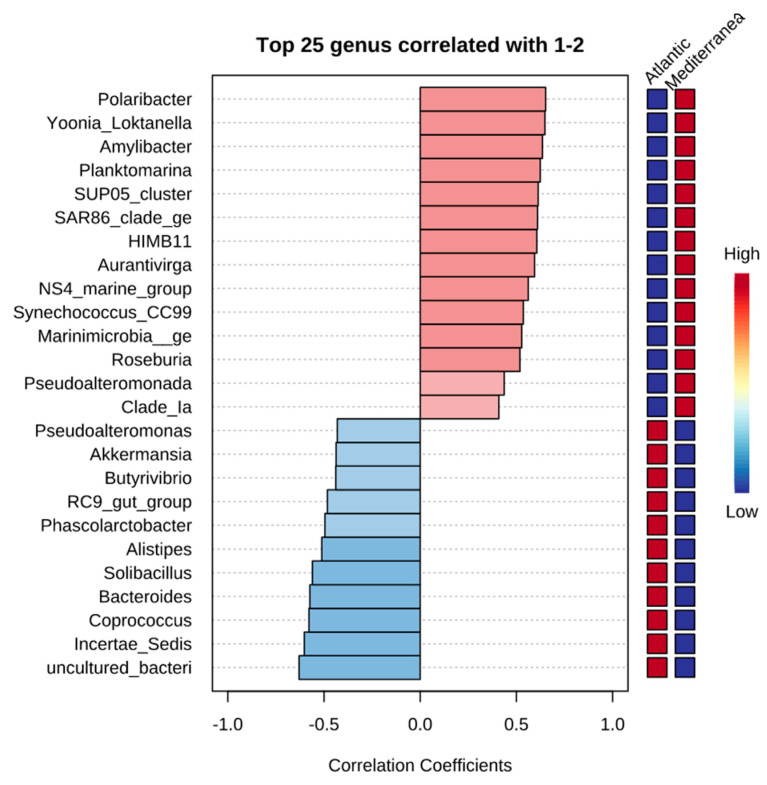
Pattern correlation analysis of the top 25 genera of the MMM. Bars indicate the value of the correlation coefficient of a significantly correlated taxon with the source at the genus level. Correlation coefficients are represented by positive (red) or negative (blue) correlations. Significant difference (*p* < 0·05).

**Figure 10 microorganisms-10-01622-f010:**
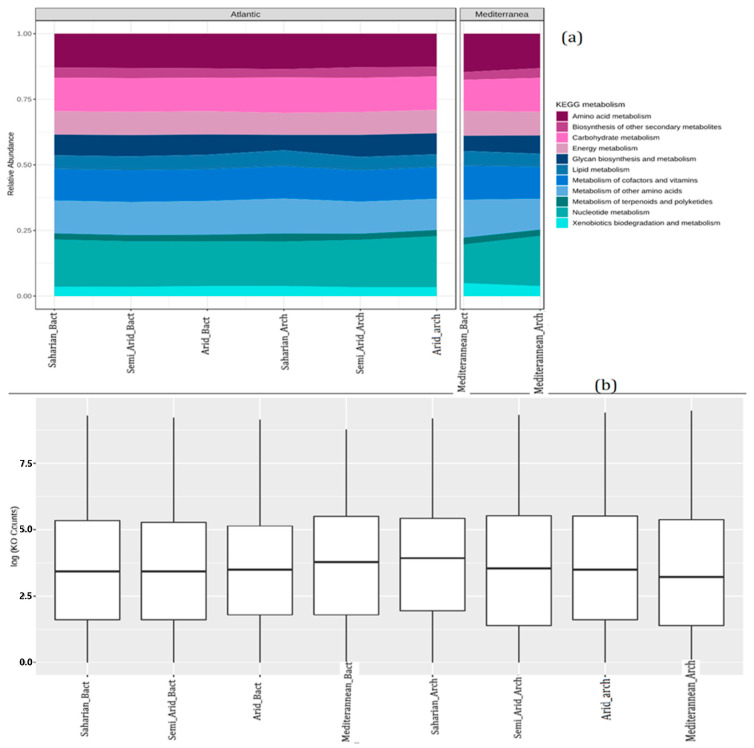
Distribution of counts in predicted metagenomic abundance data (KO counts) (logarithmic scale) (**b**). Functional profiling at the KEGG metabolic level using a stacked area graph. Samples are present on the *x*-axis and colored according to metadata (age group). The relative abundances of the various metabolic categories across all samples are plotted on the *y*-axis (**a**).

**Table 1 microorganisms-10-01622-t001:** Atmospheric particles and chemical parameters of the different climates extracted from TaraMicobiome mission.

Sample ID	Saharian	Arid	Semi-Arid	Mediterranean
Carbon Monoxide (COSC); (µg/m^3^)	138.69	140.87	144.86	147.04
Carbon Dioxide (CO_2_sc); (mg/m^3^)	774.59	775.62	780.11	787.34
Sulfur Dioxide (SO_2_sm); (µg/m^3^)	0.07	0.11	0.21	0.86
Nitrogen Dioxide (NO_2_)	No data	No data	No data	No data
Dust Extinction (Aerosol Optical Thickness, 550 nm) (Duex) (T)	0.1009	0.0044	0.0203	0.0278
Particulate Matter < 1 µm (PM1); (µg/m^3^)	1	1	1	1
Particulate Matter < 2.5 µm (PM2.5); (µg/m^3^)	6	3	4	4
Particulate Matter < 10 µm (PM10); (µg/m^3^)	21	8	9	21
Sulfate Extinction (Aerosol Optical Thickness, 550 nm) (AOT) (SO_4_ex) (T)	0.013	0.015	0.016	0.013

**Table 2 microorganisms-10-01622-t002:** Descriptive table of the amplification program of the V4 region of the 16SrRNA gene for Archaea and Bacteria.

Gene-Region	Phyla	Ref-Primer	Program		Reference
**16S-V4**	** *Bacteria* **	**F** = CS1-TS-B341F	98 °C for 3 min		[7]
			25 Cycles	95 °C for 30 s	
				55 °C for 30 s	
		**R** = CS2-TS-805NR		72 °C for 30 s	
			72 °C for 5 min		
**16S-V4**	** *Archaea* **	**F** = CS1-F-A519F	98 °C for 3 min		[8]
			25 Cycles	95 °C for 30 s	
				60 °C for 30 s	
		**R** = CS2-TS-Arch-855R		72 °C for 30 s	
			72 °C for 10 min		

**Table 3 microorganisms-10-01622-t003:** Physical and chemical in situ parameters of the different climates.

Sample ID	Saharian	Arid	Semi-Arid	Mediterranean
Temp; (°C)	19.7	20	19	17
pH	8.15	8.15	8.3	8.3
Salinity; (‰)	37.5	37.1	37.6	41.2
Dissolved oxygen; (mg/L)	9.8	7.7	10.3	9.5

**Table 4 microorganisms-10-01622-t004:** Correlations of taxa at the genus level present in the Atlantic with those in the Mediterranean.

Genus	*r*	*p*
*Polaribacter*	0.651	0.0801
*Yoonia_Lokatanella*	0.647	0.0823
*Amylibacter*	0.635	0.0905
*Planktomarina*	0.623	0.0985
*SUP05_cluster*	0.6131	0.1059
*SAR86_clade_ge*	0.609	0.1089
*HIMB11*	0.606	0.1110
*Aurantivigra*	0.594	0.1201
*NS4_marine_group*	0.561	0.1471
*Synechococcus_CC9902*	0.2646	0.0297
*Marinimicrobia_ge*	−0.27	0.0396
*Roseburia*	0.536	0.170
*Pseudoalteromonadaceae*	0.436	0.2796
*Clade_la*	0.408	0.3147
*Pseudoalteromonas*	−0.430	0.2864
*Akkermansia*	−0.437	0.2787
*Butyrivibrio*	−0.439	0.276
*RC9_gut_group*	−0.482	0.2257
Phascolarctobacter	−0.495	0.2118
*Alistipes*	−0.5116	0.1949
*Solibacillus*	−0.560	0.148
*Bacteroides*	−0.572	0.1376
*Coprococcus*	−0.577	0.1336
*Incertae_Sedis*	−0.602	0.114
*Uncultured_bacteria*	−0.630	0.0940

**Table 5 microorganisms-10-01622-t005:** Differential abundance of MMM according to Moroccan coastal climate. Wald tests of differential log2FC abundances of OTUs. Twenty-six genera revealed significant features, based on *p*-values, with differential abundance.

	log2FC	logCPM	*p*-Values	FDR
*Polaribacter*	7.6426	14.113	3.0624 × 10^−5^	0.0040577
*Yoonia_Loktanella*	6.9335	14.034	5.1361 × 10^−5^	0.0045369
*Amylibacter*	6.8861	15.819	0.00029882	0.012863
*Rhodobacteraceae_unclassified*	5.8166	12.39	0.00033977	0.012863
*SAR86_clade_ge*	5.625	13.232	0.00050154	0.016613
*Rhodobacteraceae_unclassified_5*	4.2462	10.795	0.00065675	0.018107
*Aurantivirga*	5.7707	14.009	0.00068327	0.018107
*Planktomarina*	4.9286	11.689	0.00081746	0.019693
*HIMB11*	5.2131	12.78	0.00091929	0.020301
*SUP05_cluster*	4.6165	11.331	0.0012203	0.023696
*Candidatus_Actinomarina_1*	3.9349	10.722	0.0012519	0.023696
*Synechococcus_CC9902*	3.8306	11.186	0.0019083	0.031766
*NS4_marine_group*	4.1886	11.221	0.0019179	0.031766
*SAR86_clade_ge_1*	4.4104	11.766	0.0025391	0.03958
*Marinimicrobia__ge*	4.1083	11.494	0.0034288	0.047822
*Enterobacterales_unclassified_3*	3.0121	10.025	0.0048204	0.062794
*Pseudoalteromonadaceae_unclassified_5*	2.9056	10.043	0.0049761	0.062794
*Acinetobacter*	2.6463	10.089	0.007196	0.082911
*Clade_I_unclassified*	2.5445	10.117	0.010326	0.11401
*Marine_Group_II_ge_6*	4.0622	11.844	0.011044	0.11539
*Rhodobacteraceae_unclassified_4*	3.0368	11.103	0.011321	0.11539
*Marine_Group_II_ge_10*	3.0639	10.456	0.013922	0.13664
*Sphingorhabdus*	2.4657	9.9038	0.015954	0.151
*Rhodobacteraceae_unclassified_1*	3.6972	12.13	0.017132	0.15656
*Clade_Ia*	4.1302	15.278	0.019241	0.16996
*Marine_Group_II_ge_9*	2.6756	10.447	0.029006	0.24795
*Candidatus_Puniceispirillum*	2.4428	10.843	0.029999	0.24843
*NS5_marine_group*	3.2383	12.426	0.032082	0.25763
*OM60_clade_1*	2.3647	10.552	0.036781	0.28667
*Woesearchaeia_ge_1*	2.5099	10.189	0.04154	0.31452
*Glaciecola*	3.2737	12.479	0.046068	0.3365
*Clade_II_ge*	2.7034	11.728	0.048253	0.3365
*Marine_Group_II_ge_8*	2.4898	10.685	0.053948	0.35414
*Pseudohongiella*	2.0693	9.8599	0.054751	0.35414
*Marine_Group_II_ge_7*	3.0471	11.682	0.063485	0.40056
*Candidatus_Actinomarina*	2.5996	11.999	0.071843	0.44276
*Candidatus_Nitrosopumilus_3*	2.332	10.727	0.073943	0.44534
*Woesearchaeia_ge*	2.2609	10.747	0.09075	0.5228
*Marine_Group_II_ge_3*	2.7335	12.328	0.099505	0.56104
*Marine_Group_II_ge_4*	2.6089	12.202	0.1147	0.63322
*Pseudoalteromonas_1*	−5.2091	15.565	0.13898	0.75162
*Marine_Group_II_ge_11*	1.9035	10.397	0.15393	0.81582
*Thermoplasmata_unclassified*	1.715	10.45	0.18448	0.95639
*Marine_Group_II_ge_2*	2.2069	12.934	0.18767	0.95639
*Marine_Group_II_ge_5*	2.0212	12.197	0.20278	0.98455
*Flavobacteriaceae_unclassified*	1.1644	9.7761	0.22177	1
*Pseudoalteromonas*	−4.3703	16.223	0.23593	1
*Marine_Group_II_ge_1*	2.032	15.231	0.245	1
*Marine_Group_II_ge_14*	1.4912	10.295	0.26058	1
*NS5_marine_group_1*	1.062	10.659	0.28825	1
*Candidatus_Nitrosopelagicus*	1.4629	13.514	0.3687	1
*Cobetia*	−2.7752	13.03	0.37506	1
*Nitrosopumilaceae_unclassified*	1.3204	13.01	0.39378	1
*Candidatus_Nitrosopumilus_2*	1.2182	10.994	0.39482	1
*Pseudoalteromonas_2*	−1.9949	10.992	0.42023	1
*Rhodobacteraceae_unclassified_3*	0.92903	10.463	0.42875	1
*Psychrobacter*	−2.6193	12.888	0.45942	1
*Pseudoalteromonas_10*	0.68628	9.712	0.47556	1
*Nitrosopumilaceae_unclassified_2*	0.88441	10.555	0.4964	1
*Vibrionaceae_unclassified*	−1.8098	11.329	0.51009	1
*Marine_Group_II_ge*	0.97786	16.549	0.51011	1
*Shewanella*	−1.5	10.617	0.55035	1
*Nitrosopumilaceae_unclassified_4*	0.78774	10.43	0.55367	1
*Nitrosopumilaceae_unclassified_1*	0.64499	12.682	0.59802	1
*Bacteria_unclassified_5*	−1.2119	10.405	0.61273	1

## Data Availability

Not applicable.

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
