# Peer review of "Effect of Climate on Bacterial and Archaeal Diversity of Moroccan Marine Microbiota"

_microorganisms, 2022, doi:10.3390/microorganisms10081622_

Round 1

Reviewer 1 Report

The paper entitled ‘Effect of climate on Bacterial and Archaeal diversity of Moroc-2 can Marine Microbiota’ by Yousra Sbaoui and co-authors provides insight into Moroccan marine microbiota sampled in the areas characterized by the diverse climatic conditions.

General comments:

I suggest changing the set of key words. Most of them are in abstract.

What was the scientific hypothesis checked during the research?

Did you analyze the samples of a negative control?

Specific comments:

L.19 ‘(Mediterranean, semi-arid, arid and Saharian)’ can be omitted

L.24 ‘MMMs’. What is it?

L.26 – 27 ‘Four water samples of twelve sampling sites’. What do you mean?

L.34 ‘Thaumarchaeota. identified’

L.39 – 42. Please, be more specific.

L.62 – 68 can be omitted.

L.100. ‘in-situ’. In italic.

L.130 ‘2.5.116. rRNA’ ???

I think Table 2 should be in the Section 2.1.

Please, transfer figure 2 and 10 to Supplementary.

L.230 – 232, 302 – 307, 431 – 439 can be omitted

L.232 – 238 should be in the Section 2.

L.243 – 286, 416 – 420, 440 – 449. Please, shorten.

L.313 ‘we’ve’.

L.314. Is it significant?

L.488 ‘with previous work’. ‘a’

L.523 ‘diversity.6. Patents’ ???

Author Response

Response to Reviewer 1 Comments

Point 1: I suggest changing the set of keywords. Most of them are in the abstract.

Response 1: These keywords reflect the main idea of the manuscript and are the most used throughout the sections mentioned.

Point 2: What was the scientific hypothesis checked during the research?

Response 2: Thank you for this important question.

Our main research question is: What are the biotic and abiotic parameters that act on the survival and behavior of marine microbiota in the surface microlayer, in particular the Bacterio-Neuston?

Through our research projects each time we respond to a parameter. In this article we have discussed the climate according to the altitudinal gradient with the following hypothesis:

If the authoctone marine microbiota in the surface microlayer changes according to the physicochemical parameters of the environment, then we can observe a change in its diversity, composition, function, and response along the latitudinal gradient (climate) because according to bibliography sources, the surface microlayer is the first receptacle for the majority of the environmental external factors.

Point 3: Did you analyze the samples of a negative control?

Response 3: During this study the only negative controls used, they were related to the amplification and the sequencing, but, there was no sequence to analyze (they are negative control). If you designate by your question a control site that is not affected by climate, according to the sampling map (fig.1) all sites of the Moroccan coastline were affected by the latitudinal gradient.

The third option, which is linked to a negative control during the post-sequencing analysis step, applies the analysis protocol on a sequence already published with its results to validate the results to be obtained and the different graphical representations mentioned in this manuscript. In this case yes

Point 4: L.19 ‘(Mediterranean, semi-arid, arid, and Saharian)’ can be omitted

Response 4: Done

 Point 5: L.24 ‘MMMs’. What is it?

Response 5: Typing error that we have corrected

Point 6: L.26 – 27 ‘Four water samples of twelve sampling sites’. What do you mean?

Response 6: Each site had 3 sampling points (upstream, middle, and downstream), and since we had 4 major climate types 3*4=12

Point 7: L.34 ‘Thaumarchaeota. Identified

Response 7: Done

Point 8: L.39 – 42. Please, be more specific

Response 8: Done.

Point 9: L.62 – 68 can be omitted.

Response 9: Done.

Point 10: L.100. ‘in-situ’. In italic.

Response 10: Done.

Point 11: L.100. ‘in-situ’. In italic.

Response 11: Done.

Point 12: L.130 ‘2.5.116. rRNA’ ???

Response 12: Checked.

Point 13: I think Table 2 should be in Section 2.1

Response 13: Done, we have divided the data in the table into two parts. The results of the in situ measurements in section 3 and the data extracted from the Tara microbiome mission database in section 2.

Point 14: Please, transfer figures 2 and 10 to Supplementary.

Response 14: Done for figure 2 and for figure 10, we think it will be better to keep it in the main document because it shows the results of the prediction of the metabolic functions in each climate separately.

Point 15: L.230 – 232, 302 – 307, 431 – 439 can be omitted.

Response 15: Done for others and from 431-439 This is a reminder of the purpose of the article before starting the discussion.

Point 16: L.232 – 238 should be in Section 2.

Response 16: it’s related to sequencing quality results.

Point 17: L.243 – 286, 416 – 420, 440 – 449. Please, shorten.

Response 17: Done.

Point 18: L.313 ‘we’ve’.

Response 18: Done.

Point 19: L.314. Is it significant?

Response 19: Yes

Point 20: L.488 ‘with previous work’. ‘a’

Response 20: Done

Point 21: L.523 ‘diversity.6. Patents’ ???

Response 21: Corrected

Reviewer 2 Report

In this article authors investigated the effect of climate on bacterial and archaeal diversity of Moroccan Marine Microbiota. I think the manuscript needs major revision before publication.

Some comments:

Page 1 line 24. There is no explanation of the acronym “MMMs” in the abstract.

Page 1 line 46. The introduction section should be improved. Please add more details regarding your study.

Page 4 line 130. Please fix the format of the section numbering, for example “2.5.116. rRNA Sequencing” and page 9 line 301. “3.1. Alpha and beta diversity”

I think that Figures 2 and 9 are not mentioned in the MS.

All references should be checked for correctness. I got surprised by the fact that references are both in alphabetical and numerical order..      

Author Response

Response to Reviewer 2 Comments

Point 1: Page 1 line 24. There is no explanation of the acronym “MMMs” in the abstract

Response 1: We intended to write MMM as an abbreviation of Moroccan Marine Microbiota, so it is just a typing error that we corrected in the manuscript.

Point 2: Page 1 line 46. The introduction section should be improved. Please add more details regarding your study.

Response 2: Concerning this point, we have added some details on the methodology without too much detail. The goal is to keep the speçfcity to the other manuscript sections.

 Point 3: Page 4 line 130. Please fix the format of the section numbering, for example, “2.5.116. rRNA Sequencing” and page 9 line 301. “3.1. Alpha and beta diversity”

Response 3: Thank you for your comment. It's done.

Point 4: I think that Figures 2 and 9 are not mentioned in the MS

 Response 4: if you designate by MS (Supplementary Material), these 2 figures are linked to the main manuscript and are already quoted in their paragraphs.

Point 5: All references should be checked for correctness. I got surprised by the fact that references are both in alphabetical and numerical order.

Response 5: You are right. They are checked and corrected according to the recommendations of the journal. Thank you

Round 2

Reviewer 2 Report

The authors have revised their manuscript acknowledging my previous commentsthe manuscript is now ready for publication.